**Data Availability Statement:** The anonymized individual demographic and colon length data are available from the Nottingham Research Data

## Colon length in pediatric health and constipation measured using magnetic resonance imaging and three dimensional skeletonization

**Hayfa Sharif**[1,2,3], **Caroline L. Hoad**[2,4], **Nichola Abrehart**[1,2], **Penny A. Gowland**[4], **Robin C. Spiller**[1,2], **Sian Kirkham**[5], **Sabarinathan Loganathan**[5], **Michalis Papadopoulos**[5,6], **Marc A. Benninga**[7], **David Devadason**[5], **Luca Marciani**[1,2]*

1 Nottingham Digestive Diseases Centre, Translational Medical Sciences, School of Medicine, University of Nottingham, Nottingham, United Kingdom, 2 NIHR Nottingham Biomedical Research Centre, Nottingham University Hospitals NHS Trust, Nottingham, United Kingdom, 3 Amiri Hospital, Ministry of Health, Civil Service Commission, Kuwait City, Kuwait, 4 Sir Peter Mansfield Imaging Centre, School of Physics and Astronomy, University of Nottingham, Nottingham, United Kingdom, 5 Nottingham Children's Hospital, Nottingham University Hospitals NHS Trust, Nottingham, United Kingdom, 6 Evelina Children's Hospital, London, United Kingdom, 7 Emma Children's Hospital, Department of Pediatric Gastroenterology, Amsterdam UMC, Amsterdam, the Netherlands

* Luca.Marciani@nottingham.ac.uk

## Abstract

Recent magnetic resonance imaging (MRI) studies showed that colonic volumes in children are different between health and functional constipation. The length of the colon has however been rarely measured and principally using unphysiological colon preparations or cadaver studies. The main objective of this study was to measure the length of the undisturbed colon in children with functional constipation (FC) and healthy controls. Here, the colon of 19 healthy controls (10–18 years old) and 16 children with FC (7–18 years old) was imaged using MRI. Different regions of the colon (ascending, transverse, descending, and sigmoid-rectum) were first segmented manually on the MRI images. Three-dimensional skeletonization image analysis methods were then used to reduce the regions of interest to a central, measurable line. Total colon length (corrected for body surface area) in healthy controls was 56±2 cm/m$^2$ (mean±SEM). Total colon length was significantly longer in children with FC 69±3 cm/m$^2$ compared to controls (p = 0.0037). The colon regions showing the largest differences between groups were the ascending colon (p = 0.0479) and the sigmoid-rectum (p = 0.0003). In a linear regression model, there was a positive significant correlation between total colon length and age (R = 0.45, p = 0.0064), height (R = 0.49, p = 0.0031), weight (R = 0.46, p = 0.0059) and colon volume (R = 0.4543, p = 0.0061). Our findings showed significant differences in colon lengths between healthy controls and children with constipation. A new objective diagnostic imaging endpoint such as colon length may help to improve knowledge of colon morphology and function and, in turn, understanding of colon functional pathology.

Management Repository (https://rdmc.nottingham.ac.uk/) with DOI 10.17639/nott.7327.

**Funding:** This work was funded by two National Institute for Health Research (NIHR) Invention for Innovation (i4i) grants awarded to L.M., awards number II-LB-0814-20002 and NIHR200014. The views expressed are those of the authors and not necessarily those of the National Health Service (NHS), the NIHR, or the Department of Health & Social Care. H.S. was funded by an academic scholarship from the Ministry Of Health, Civil Service Commission, Kuwait. The funders did not play any role in the study design, data collection and analysis, decision to publish, or preparation of the manuscript.

**Competing interests:** The authors have declared that no competing interests exist.

# Introduction

In childhood, functional constipation (FC) can affect severely quality of life, schooling and social relations [1]. Notwithstanding a prevalence of around 10% [1], the causes of FC are not well understood [2] and diagnosis relies mostly on reported symptoms [3–5]. Objectively measured parameters of colon morphology could aid understanding and help to guide treatment. It could also aid to develop hydrodynamic modelling of the colon mass flow and drug dissolution modelling.

A limited amount of information on colon length in infants, children, and adults is available, and this originates mostly from techniques which are invasive or require colon cleansing and use of luminal contrast media such as surgical intervention [6], barium enema [7], cadavers study [8] or computed tomography (CT) [9].

There is an obvious benefit in using non-invasive imaging methods to capture the undisturbed physiological length of the colon. Magnetic resonance imaging (MRI) with its excellent tissue contrast and lack of ionizing radiation would be an ideal technique to use. A recent study by Mark et al used MRI in conjunction with a new electromagnetic 3D-Transit capsule technique to measure the length of the colon in healthy adults [10]. Total colon length was measured from the MRI images using a 3D topological skeleton algorithm in Matlab® [10] and the value had an average of 95 cm. To our knowledge this is the only study available using MRI and it was performed in adult healthy participants.

This study aimed to measure colonic length in children with FC and healthy controls using MRI and 3D skeletonization image analysis methods. We hypothesized that the colon length would be longer in patients with FC compared to healthy controls.

# Methods

## Participants and study design

The colon MRI images used for this work were collected during a previous study investigating whole gut transit time (WGTT) and colon volumes in children without and with FC [11, 12]. Briefly, that study comprised 19 healthy volunteers with no history of gastrointestinal disease (8 male; 11 female; age 16±2 years old (mean±SEM); body mass index (BMI) 25±5 kg/m$^2$) and 16 patients with functional constipation fulfilling the Rome IV criteria (7 male; 9 female; age 11±3 years old; BMI 25±9 kg/m$^2$). Individual demographic characteristics are provided in supplemental information S1 Table. The Rome IV criteria [4] were used to identify patients with pediatric FC following a referral either from primary or secondary care into a specialist clinic. The colon of the participants was in a physiological state and no bowel cleansing or preparation, nor anesthesia was used. The colon was imaged using a 3 Tesla MRI scanner (Philips, Best, the Netherlands) [11, 12]. The participants were imaged in the supine position using short breath-hold multiple-echo (mDIXON) sequence [13] (flip angle = 20°, echo time 1 = 1.32 ms, echo time 2 = 2.2 ms, repetition time = 10 ms, acquired resolution = 1.8×1.8×4.4 mm$^3$, field of view = 250x350 mm$^3$).

The UK National Research Ethics Committee approved this study (approval number 17/WM/0049). The study was also approved by the UK Medicines and Healthcare products Regulatory Agency (MHRA) with approval number CI/2017/0054. The study was prospectively registered on Clinicaltrials.gov with study identifier NCT03564249. The young participants and their carer/parents gave informed written assent or informed written consent as applicable. This included use of the data for further research. The data from the original study was accessed retrospectively between the 4$^{th}$ of March 2022 and the 5$^{th}$ October 2022 for the

purpose of this study. The MRI image data was anonymized and the authors could not identify the participants during or after data collection.

The primary outcome of this study was total colon length as measured from MRI images using 3D image analysis skeletonization methods described below. The measurement of the length of different anatomical colon regions (ascending, transverse, descending and sigmoid-rectum) were secondary outcomes. Possible correlations of total colon length with age, sex, colon volumes and WGTT were exploratory outcomes of the study.

### Image analysis

First the colon was segmented manually on each MRI slice using Medical Image Processing, Analysis and Visualization software (MIPAV, NIH, Bethesda) as described previously [12]. The colon was divided into 4 anatomical regions. The ascending colon (AC) was outlined from the cecum to the superior point of the hepatic flexure. The transverse colon (TC) was outlined from the superior point of the hepatic flexure to the splenic one. The descending colon (DC) was identified from the splenic flexure to the start of the sigmoid colon. This was defined where the descending colon deviated posteriorly or medially. The sigmoid colon and rectum region (SC-R) was outlined from the end of the descending colon to the anorectum. These regions are shown for one patient participant in Fig 1A. Next, binary masks of each segmented colon region were formed. Fig 1B shows a 2D projection of the overall 3D binary mask for the whole colon.

Colon lengths were then measured using the AnalyzeSkeleton plug-in [14] in ImageJ software (National Institutes of Health, Bethesda, Maryland, USA). Skeletonization is an image analysis process that used binary thinning to reduce three-dimensional anatomical structures to a central line (skeleton) which preserves the length connectivity in the original object whilst erasing all other foreground pixels (Fig 1C). The program then tagged all voxels in the skeleton image and counted all its junctions, triple and quadruple points and branches, and measured the average and maximum length of the individual segments. Care was taken when colon loops were particularly tortuous as the algorithm could sometime 'jump' between segments thereby underestimating the total length. Visual inspection of all segments and, if necessary, careful re-segmentation were used to ensure the correct path was traced. After the 3D skeletonization process the plug-in Analyze Skeleton2D/3D was applied to quantify the length of the colon regions.

When required by comparisons the colon length data was corrected for body surface area (BSA, in units of $m^2$) calculated using the Mosteller formula as the square root of (height (expressed in cm)×weight (expressed in kg)/3600) [12, 15, 16]. When used, the BSA correction is clearly indicated in the Results section, all other data are otherwise uncorrected.

### Statistics

Statistical analysis was carried out using Prism 9 (GraphPad Software Inc., La Jolla, CA, USA). The Shapiro-Wilk normality test was used to test the normality of the data. Two-tailed t-test or Mann–Whitney rank sum test for unpaired data were used to test differences between groups, which were considered significant at $p < 0.05$. Linear regression was used to assess the strength of the relationship between colon length and colon volume and WGTT respectively.

### Results

It was possible to carry out the 3D skeletonization analysis for all participants and all colon segments. In three participants the recto-sigmoid region was more challenging than the other

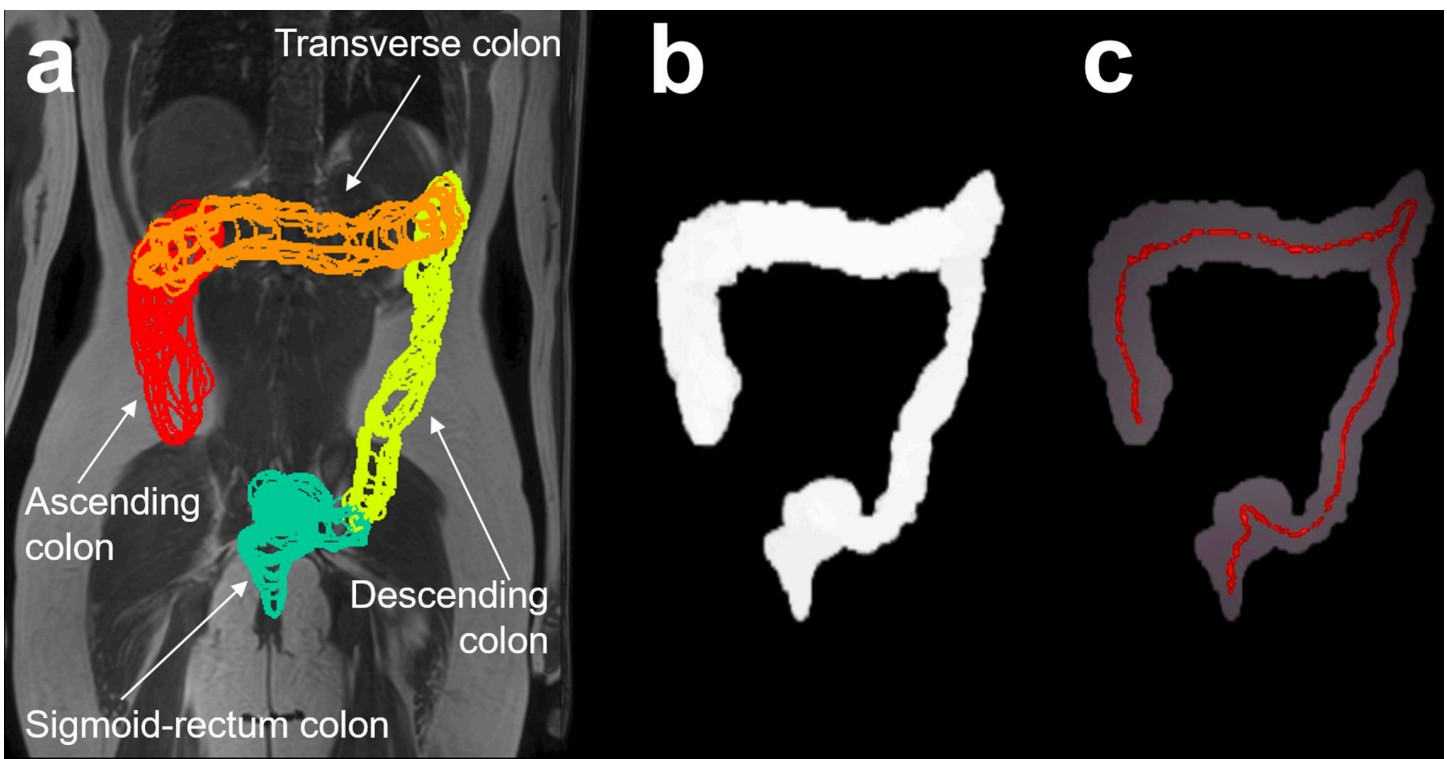

**Fig 1. Example of colon segmentation and colon length path for one patient participant.** (a) shows a 2D projection of the 3D segmentation of the 4 regions of the colon of the patient participant, superimposed to a coronal MRI scan of their body. The regions are color coded as follows: ascending colon in red, transverse colon in orange, descending colon in yellow and sigmoid/rectum colon in green. (b) shows a 2D projection of the overall 3D binary mask for the whole colon whereby the colon region is set to white and everything else is set to black. (c) shows the final 2D projection of the 3D colon length path line (shown in red) for this patient participant superimposed to the corresponding shaded colon projection outline.

regions due to its tortuosity and required re-segmentation of that colon region prior to application of the skeletonization process. The data for colon lengths are reported in Table 1.

Total colon length (not corrected for BSA) was 7% longer for the FC group compared to healthy controls but this difference was not significant. Total colon length for all participants correlated significantly with their age (R = 0.45, p = 0.0064), height (R = 0.49, p = 0.0031) and weight (R = 0.46), p = 0.0059) but not with their BMI (R = 0.17, p = 0.3265).

The total colon length corrected for BSA (Fig 2) was 55±2 cm/m$^2$ (mean±SEM) for the healthy controls. Total colon length corrected for BSA was significantly longer (69±3 cm/m$^2$) for the children with FC compared to the controls (p = 0.0037 Mann-Whitney two-tailed test).

**Table 1. Length of colon segments for 19 healthy controls and 16 patients with functional constipation (FC).** The data are shown both with actual values and also after correction for Body Surface Area (BSA). Values are mean±SEM.

| Colon segment | Length for healthy controls (cm) | Length for patients with FC (cm) | Length for healthy controls corrected for BSA (cm/m$^2$) | Length for patients with FC corrected for BSA (cm/m$^2$) |
|---|---|---|---|---|
| Ascending colon | 19±1 | 17±1 | 11±1 | 13±1 |
| Transverse Colon | 27±1 | 24±2 | 16±1 | 18±1 |
| Descending colon | 31±2 | 26±1 | 18±1 | 20±1 |
| Sigmoid-rectum colon | 20±1 | 22±2 | 11±1 | 17±1 |
| Total colon | 96±3 | 90±5 | 56±2 | 69±3 |

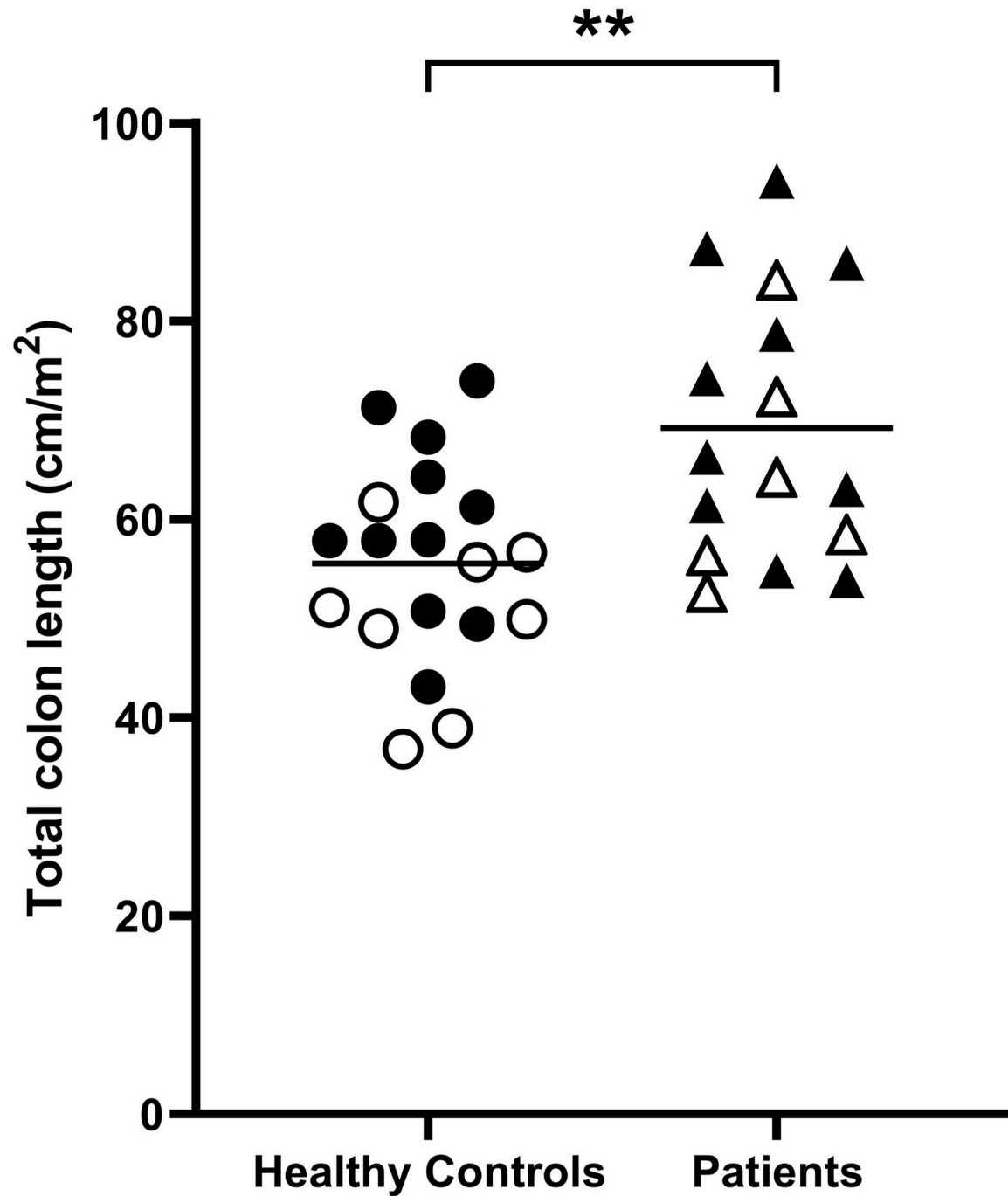

**Fig 2. Total colon length corrected for both body surface area of n = 19 control and n = 16 patients with functional constipation.**
Solid symbols indicate female participants and open symbols indicate male participants respectively. The horizontal lines indicate the mean, ** p = 0.0037 two-tailed Mann-Whitney test.

The total colon length corrected for BSA was significantly longer for female controls 60±10 cm/m² than for male controls 50±9 cm/m² (p = 0.0361 unpaired two tailed t test). It was also longer for female FC patients 74±5 cm/m² than for male FC patients 63±4 cm/m², but the difference was not significant (p = 0.1119 unpaired two tailed t test). There was no correlation

between age and actual (uncorrected for BSA) total colon length in the control group (p = 0.9759), whilst these correlated strongly in the constipation group (R = 0.68, p = 0.0039).

Regional colon lengths, corrected for BSA, are shown in Fig 3. The AC and SC-R regions were shorter in the control group compared to the FC group, p = 0.0479 and p = 0.0003 respectively (Mann-Whitney one-tailed test).

There was a positive, significant correlation between total colon length and total colon volume (data from [12]) in the healthy controls, coefficient of correlation R = 0.56, p = 0.0132. There was also a positive, significant correlation between total colon volume and total colon length in the patients with FC, coefficient of correlation R = 0.63, p = 0.0093. No correlation between total colon length and whole gut transit time (data from [11]) was found in the control group (p = 0.5333) nor in the patients' group (p = 0.2582) .

## Discussion

In this study we successfully used a 3D skeletonization method to measure the length of the colon from MRI images in a group of healthy children and a group of children with functional constipation (FC). These data were collected in a physiological state, not disturbed by purges or oral/enteral contrast media (liquid or gas). To the best of our knowledge, we are not aware of similar pediatric measurements being available in the literature.

The limited data on human colon length in the literature, and particularly the fact that previous studies generally disturbed colon physiology with cleaning preparations and/or luminal contrast media, make it particularly difficult to put our results in context with other studies. Colonic elongation was found in 23% of children with slow transit constipation [17]. This was defined as colon contour appearance on gamma scintigraphy being unusually long or tortuous, and 30% longer than normal appearance. The number of colon redundancies (as a marker of elongated colon length) in functional fecal retention were also found to correlate significantly with colon transit time and symptoms [18]. These studies however did not measure actual colon lengths, which our work measured. As we hypothesized, colon length was longer in the group of children with functional constipation than in the controls, showing that the colon of these children not only has a larger volume as previously found [12], but also a greater length. This illustrates that the organ volume changes shown in [12] are actually reflected in elongation which, in turn, may affect longitudinal muscle physiology. Colon elongation, affecting the longitudinal muscles, may be secondary as a result of fecal impaction [19], though it is difficult to dissect whether colonic elongation might be a primary factor in constipation [17]. The colon in females was longer than in males in keeping with previous findings [19]. The reasons for this difference are unclear but could include hormonal and/or gynecological factors [19].

Colon length correlated with age for the FC patients but not in the healthy controls. The control group was older than the FC group and could have presented with a more developed colon size, but to what extent this could have influenced the correlation is unclear. A previous study did not find a correlation with age although the colon in that study was inflated with air contrast enemas and the children were younger than in our study [20]. In the retrospective computed tomography study by Mirjalili [9] the colon length increased from 52 cm in children between 0 and 2 years of age to 95 cm in children aged 9–11. This compares to adult values reported by others between 108 and 132 cm. A large computed tomography study reported the total adult colon length to be 189.5±23.6 cm [21].

In our study the large differences in size of the participants and the significant correlations of colon length with height and weight of the participants justified the use of the Mosteller formula to correct for BSA to allow for more meaningful comparisons between groups [12, 15, 16]. Computed tomography studies in adults did not find a correlation of colon length with

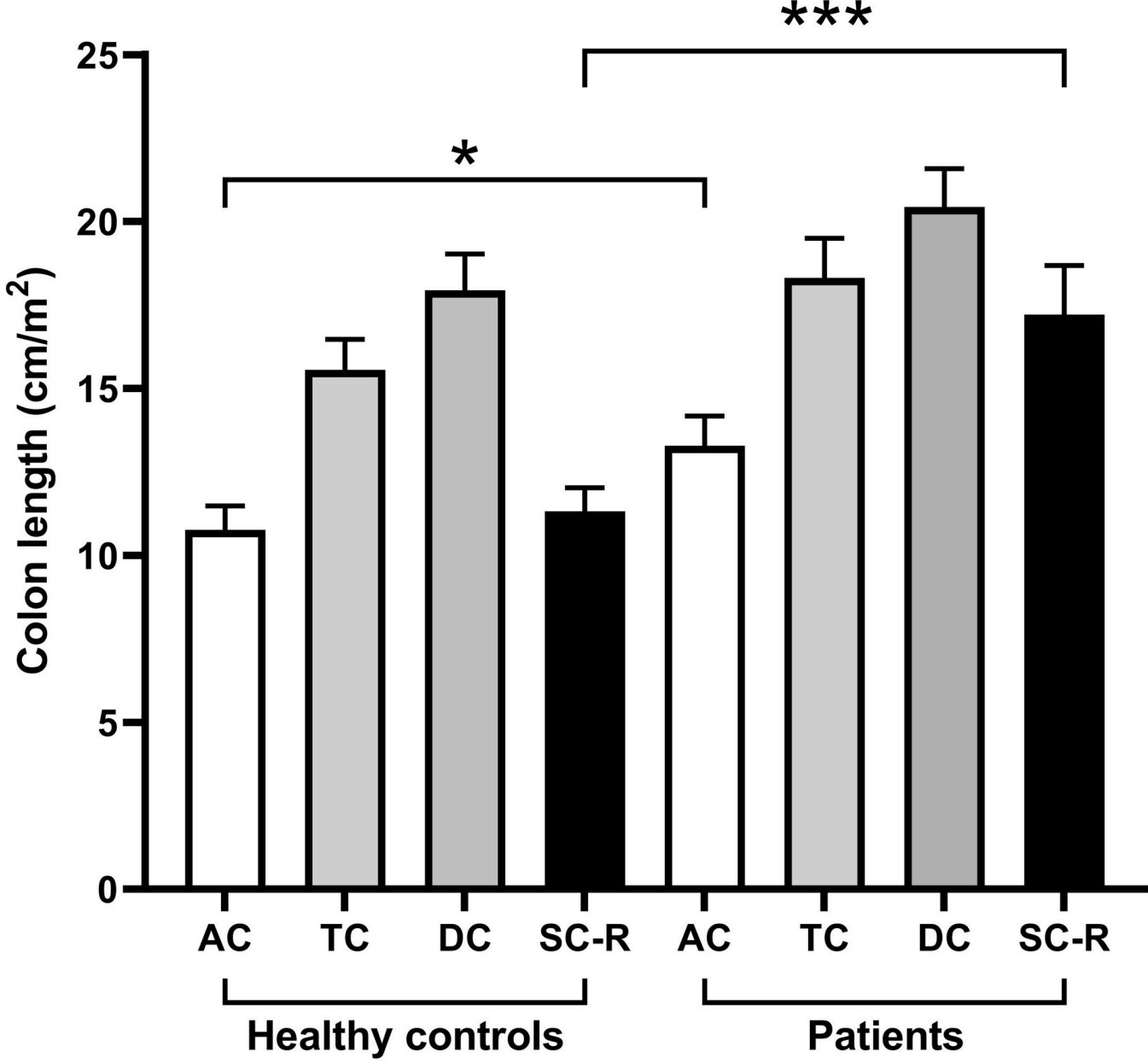

**Fig 3. Regional colon length corrected for body surface area for 19 controls and 16 patients with functional constipation.** The data are shown as mean ±SEM for the ascending colon [AC], transverse colon [TC], descending colon [DC] and sigmoid colon and rectum [SC-R]. * p = 0.0479, *** p = 0.0003 one-tailed Mann-Whitney test.

height [21]. Our findings in children correlating colon length with height and weight but not with WGTT are based on a relatively small sample and will require further investigation.

The regional segment lengths were exploratory outcomes and the p values here were not corrected for multiple comparisons. Correction for multiple comparison would still show that the SR-C differences were significant. Low numbers mean that some possible correlations with sex were difficult to explore. This study showed clear differences in colon length between the young patients with FC and the healthy controls. The largest difference in colon length was observed in the SC-R region, followed by the difference in AC region. Fecal impaction occurs

predominantly in the SC-R region whilst the AC accommodates input from the small bowel. Inferences on the relevance of these findings on the pathogenesis of FC could however not be drawn from these data only. Future studies will need to measure colonic length in patients before and after treatment to start investigating whether colon elongation could be a contributing cause of FC or secondary to prolonged fecal retention. It has been reported that the age of puberty can coincide with the time when decreased bowel frequency and other symptoms are first noticed [22], but in this study data on the age of puberty were not collected so it is not possible to comment on colon length differences before and after puberty. Future studies should also specifically stratify by pubertal status. Colonic length could also be assessed dynamically after prokinetic and antispasmodic drugs which may well act differently on longitudinal versus circular muscle and hence alter length / volume ratios. The degree of colon ptosis, as observed on X-ray images, has been linked to the severity of slow transit constipation [23]. In this study we did not observe substantial colon ptosis and further studies with larger groups of patients will be able to investigate this observation further.

MRI uses radio waves and is safe to use in children, providing they do not have incompatible metal implants in their body. Other medical imaging techniques such as computed tomography deliver a considerable dose of ionizing radiation, making them unsuitable for this purpose.

The study had some limitations. Attempts to calculate the whole colon length in one single skeletonization step were impaired by the fact that sometimes the binary masks had a missing connection between segments in the representation of the entire colon due to image quality or lack of continuity between some image slices of the colon in the segmentation. Therefore, the colon length measurements were carried out in the binary masks for each individual segmented colon region and then the total colon length were measured by summing up the regional colon length values.

The colon length analysis technique works well, providing that the MRI images are of sufficient quality and not particularly affected by respiratory motion, which was the case for this study. The technique was relatively easy to apply in most of the colon regions, but the method was somehow more difficult to apply in the rectosigmoid region due to the complexity of the rectosigmoid colon shape. In particularly tortuous loops, the algorithm can easily 'jump' between loops thereby underestimating the length of the segments. Because of this, visual inspection of every 3D skeleton line was necessary and was carried out for all data sets. In fact, in this study 3 of the recto-sigmoid regions showed 'jumps' between loops and required more careful re-segmentation to restore the correct skeletonized path. Conversely, the separate segmentation of the four anatomical segments prevented the skeletonized midline from continuing seamlessly at the junction between different segments which overestimated the length by a small amount.

The original study [11] excluded children below the age of 7 years old. This was because children younger than 7 may have only limited experience/ability of swallowing tablets and may have reduced compliance with the MRI procedures. As such, the colon length findings of this study are applicable only to children above the age of 7 years old.

## Conclusion

This study describes novel measurements of colonic length in healthy children and children with functional constipation, using MRI to image the colon not disturbed by unphysiological bowel preparations and 3D skeletonization to measure organ length. Whilst providing new data that will help improve our understanding of colon pathophysiology and drug release modelling

in children, the study also highlighted some fundamental limitations of the skeletonization technique that will need addressing to make the method more applicable in the future.

## Supporting information

**S1 Table. Demographic characteristics of the participants.** Age, sex, height, weight and body mass index (BMI) for the participants in the functional constipation patient group and the healthy control group.
(PDF)

## Author Contributions

**Conceptualization:** Hayfa Sharif, Luca Marciani.

**Data curation:** Hayfa Sharif.

**Formal analysis:** Hayfa Sharif.

**Funding acquisition:** Luca Marciani.

**Investigation:** Hayfa Sharif, Nichola Abrehart, Sian Kirkham, Sabarinathan Loganathan, Michalis Papadopoulos, David Devadason, Luca Marciani.

**Methodology:** Hayfa Sharif, Caroline L. Hoad, Luca Marciani.

**Project administration:** Luca Marciani.

**Supervision:** Luca Marciani.

**Writing – original draft:** Hayfa Sharif, Luca Marciani.

**Writing – review & editing:** Hayfa Sharif, Caroline L. Hoad, Nichola Abrehart, Penny A. Gowland, Robin C. Spiller, Sian Kirkham, Sabarinathan Loganathan, Michalis Papadopoulos, Marc A. Benninga, David Devadason, Luca Marciani.

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
