## [Decision Letter · Decision Letter 0]

16 Oct 2023

PONE-D-23-27259Colon length in pediatric health and constipation measured using magnetic resonance imaging and three dimensional skeletonizationPLOS ONE

Dear Dr. Marciani,

Thank you for submitting your manuscript to PLOS ONE. After careful consideration, we feel that it has merit but does not fully meet PLOS ONE’s publication criteria as it currently stands. Therefore, we invite you to submit a revised version of the manuscript that addresses the points raised during the review process.

We look forward to receiving your revised manuscript.

Kind regards,

Tsutomu Kumamoto

Academic Editor

PLOS ONE

Journal Requirements:

Colonic Volume Changes in Paediatric Constipation Compared to Normal Values Measured Using MRI - https://doi.org/10.3390/diagnostics11060974

3. In your revision ensure you cite all your sources (including your own works), and quote or rephrase any duplicated text outside the methods section. Further consideration is dependent on these concerns being addressed.

Additional Editor Comments:

The authors presented colon length in pediatric health and constipation using MRI. This study is a very interesting, but I have a few questions.

Specific comments

1. The authors indicated the colon length for all participants correlated significantly with their age, height and weight. Specifically, to what age do the results of this study apply to children? Would it be a good idea to include data on age and gender etc.. as a supplemental table?

2. I comprehended the largest differences between groups were the ascending colon and the recto-sigmoid colon. What do these results mean?

3. MRI was used in this study. Is there any difference or advantage between MRI and other tests such as intravenous or CT?

Reviewers' comments:

Reviewer's Responses to Questions

**Comments to the Author**

1. Is the manuscript technically sound, and do the data support the conclusions?

Reviewer #1: Yes

Reviewer #2: Yes

2. Has the statistical analysis been performed appropriately and rigorously? 

Reviewer #1: Yes

Reviewer #2: Yes

3. Have the authors made all data underlying the findings in their manuscript fully available?

Reviewer #1: Yes

Reviewer #2: Yes

4. Is the manuscript presented in an intelligible fashion and written in standard English?

Reviewer #1: Yes

Reviewer #2: Yes

5. Review Comments to the Author

Reviewer #1: This is an interesting paper that introduces a novel and simple approach to measure colonic length using non-radioactive measurements (MRI). They measure some simple colon length measures using a skeletonization algorithm.

In my opinion, the manuscript only needs minor editing before it is suitable for publication.

In general, check the punctuations, there are some places where there is a space too much between e.g., the p-value and the “=” sign. For example, in line 177, 189, 203, 205. These needs to be consistent and in line with the journal specifications.

Line 128, figure caption. There is a “is” to much.

Line 162, “are” instead of “and”

Line 234, “thank” instead of “than”

Line 256, “with sex and ..” missing a word

Abstract:

- You only mention the age of the constipated children. Can you add that the control group is age-matched or provide their range of age also?

Methods:

- To my knowledge of skeleton algorithms (I have not tried the plugin you have used), it can be difficult to avoid that the colon midline does not take “shortcuts” between the colon segments in the cases, where the segmented volumes are very close to each other, for example in the flexures or in the sigmoid area. This will potentially underestimate the true length measure. If you took any precautions to avoid this, then please describe it so others can easier replicate the work.

o You have described some of the precautions in the discussion. It would be simpler to get the relevant information in the methods section.

o When doing the skeleton algorithm on the different segments one at a time, then the midline will end at the top/bottom of the segment and not at the middle of the segment/junction to the next segment. Do the used algorithm take this into account, i.e., using a predefined start/end point. Or else each segment might be a little overestimated. Can you elaborate a little bit more on the analysis steps, so there is no doubt on how this works. I would like to see figure 1c from a sagittal view also.

Results

- You state that the hypothesis is about colon length differences between the groups. That result should be mentioned before the BSA corrected values. One might argue that the BSA corrected values are most representative, however you should be truer to your initial hypothesis.

Discussion:

- Can you include your opinion on measures of colon ptosis in relation to your total colon or transverse colon estimation? A very recent study has shown the correlation between slow transit constipation and transverse colon ptosis as measured on abdominal x-rays: Sun et al; Degree of transverse colon ptosis: an alternative surrogate for evaluation of slow transit constipation. Quantitative Imaging in Medicine and Surgery; Vol 13, No 9 (September 01, 2023)

- In line 232 you mention that colon length was correlated with age in the FC group only. Could that be due to the fact that the control group was older than the FC group, meaning that many of the controls might have reached the full-grown length of the colon, why a correlation only exists when you are still growing?

Reviewer #2: Comments on this study:

1. The authors should provide information concerning the reliability of MRI images

using 3D image analysis skeletonization technique in measurement of colon length.

2. It is well accepted that MRI study has a limitation in young children who are not cooperative with this time consuming technique. It is noted that the children enrolled in this study aged over 10 years, therefore, the data of this study would be applicable to only older children , not children in general.

3. Data in this study shows that the colon length of constipated children is longer than healthy children. However, the results do not clearly indicate that colonic length contributes to the pathogenesis of functional constipation. To investigate whether the longer colon is a cause of functional constipation or secondary to prolonged fecal retention , a study should be designed to compare colonic length measured by MRI studies in the same patient at pre treatment and at recovery post treatment of functional constipation.

6. PLOS authors have the option to publish the peer review history of their article (what does this mean?). If published, this will include your full peer review and any attached files.

Reviewer #1: **Yes: **Esben Bolvig Mark

Reviewer #2: No

---

## [Author Response · Author response to Decision Letter 0]

25 Nov 2023

PONE-D-23-27259 Revision R1

Colon length in pediatric health and constipation measured using magnetic resonance imaging and three dimensional skeletonization

Response to Reviewers

We are very grateful for the comments received from the Academic Editor and from the two Reviewers. We have addressed all the points raised as detailed below in this point-by-point response. All changes made to the manuscript are highlighted in yellow in the tracked changes version enclosed. 

We believe that the changes made in response to the reviews have indeed improved the manuscript and we hope that it would meet now the journal’s criteria.

Editorial Comments:

Answer: We checked the formatting examples and we used the template provided

Colonic Volume Changes in Paediatric Constipation Compared to Normal Values Measured Using MRI - https://doi.org/10.3390/diagnostics11060974

3. In your revision ensure you cite all your sources (including your own works), and quote or rephrase any duplicated text outside the methods section. Further consideration is dependent on these concerns being addressed.

Answer: Apologies for the similarities with the previous paper, we have revised in the Introduction and Methods Lines 58-62, Lines 76-77, Lines 92-94, Lines 95-100, Lines 112-118 and Lines 154-156 to avoid overlap. The source of the original data for this work is clearly referenced.

Answer: we checked the references list and only added the one suggested by a Reviewer below.

Additional Editor Comments:

The authors presented colon length in pediatric health and constipation using MRI. This study is a very interesting, but I have a few questions.

Specific comments

1. The authors indicated the colon length for all participants correlated significantly with their age, height and weight. Specifically, to what age do the results of this study apply to children? Would it be a good idea to include data on age and gender etc.. as a supplemental table?

Answer: In response to this and also Reviewer 2 comments we have specified more clearly what is the age range and that there was a 7 years (and older) cut-off in the inclusion criteria and why at the end of Discussion Lines 294-297. As suggested, we have also added a supplemental table S1 with the demographic details.

2. I comprehended the largest differences between groups were the ascending colon and the recto-sigmoid colon. What do these results mean?

Answer: Thank you for this comment, in response to this and also to a point made by Reviewer 2 we have added to the discussion at Lines 274-281, to say that these regions have functional and pathological relevance and that from these data only one cannot derive whether the length differences are an effect of being constipated or a cause of constipation.

3. MRI was used in this study. Is there any difference or advantage between MRI and other tests such as intravenous or CT?

Answer: Thank you for this comment, we have added a comment in Discussion Lines 291-294 to make the point that MRI uses radio waves and is safe to use in children, providing they do not have incompatible metal implants in their body. Other medical imaging techniques such as computed tomography deliver a considerable dose of ionizing radiation, making them unsuitable for this purpose.

General comments to the Author

We are very grateful for the positive appraisals of technical soundness, conclusions supported by data, appropriate and rigorous statistics, data availability and manuscript presentation. Thank you.

Review Comments to the Author: Reviewer #1

This is an interesting paper that introduces a novel and simple approach to measure colonic length using non-radioactive measurements (MRI). They measure some simple colon length measures using a skeletonization algorithm. In my opinion, the manuscript only needs minor editing before it is suitable for publication.

Answer: Thank you for the very positive appraisal of our manuscript.

In general, check the punctuations, there are some places where there is a space too much between e.g., the p-value and the “=” sign. For example, in line 177, 189, 203, 205. These needs to be consistent and in line with the journal specifications. Line 128, figure caption. There is a “is” to much. Line 162, “are” instead of “and” Line 234, “thank” instead of “than” Line 256, “with sex and ..” missing a word

Answer: thank you so much, apologies for the imperfections, we have looked out for those and run a new full spelling and grammar check.

Abstract:

You only mention the age of the constipated children. Can you add that the control group is age-matched or provide their range of age also?

Answer: agreed, we provide now the specific age range of the control group (10-18 years old) in the Abstract Line 40.

Methods:

- To my knowledge of skeleton algorithms (I have not tried the plugin you have used), it can be difficult to avoid that the colon midline does not take “shortcuts” between the colon segments in the cases, where the segmented volumes are very close to each other, for example in the flexures or in the sigmoid area. This will potentially underestimate the true length measure. If you took any precautions to avoid this, then please describe it so others can easier replicate the work.

Answer: thank you for this comment, we agree that our description in Discussion was not making this clear. In answer to this, and to a similar point raised by Reviewer #2, we have added in Discussion a section to this effect at Lines 259-267. In that section we mention the possibility that the algorithm would ‘jump’ between loops and that visual inspection is necessary. We also reinforce what was already said at the beginning of Results, that in 3 cases careful re-segmentation of the sigmoid-rectum region was necessary to prevent the ‘jumps’. We point out now that ‘jumping’ would underestimate the segment length measured. 

o You have described some of the precautions in the discussion. It would be simpler to get the relevant information in the methods section.

Answer: agreed we have added a description of the precaution in the Methods at lines 142-145 and also left some discussion of this in the Discussion section. 

o When doing the skeleton algorithm on the different segments one at a time, then the midline will end at the top/bottom of the segment and not at the middle of the segment/junction to the next segment. Do the used algorithm take this into account, i.e., using a predefined start/end point. Or else each segment might be a little overestimated. Can you elaborate a little bit more on the analysis steps, so there is no doubt on how this works. 

Answer: agreed, the algorithm did not allow to predefine start and stop points in the binary masks so you are correct, when the separate segments were summed up, this will have overestimated the total length by a small amount. We have now inserted a comment to this effect to make the point clear in Discussion lines 267-270. 

I would like to see figure 1c from a sagittal view also.

Answer: we tried to do this at least for one example figure to answer your request positively, but the manual segmentation of the colon is done on multi-slice coronal images with larger slice resolution than in-plane, and when trying to reconstruct in a different orientation the quality of the resulting images and projections is not very good. Apologies for not satisfying this one request. 

Results

- You state that the hypothesis is about colon length differences between the groups. That result should be mentioned before the BSA corrected values. One might argue that the BSA corrected values are most representative, however you should be truer to your initial hypothesis.

Answer: Agreed, we have now added this first in Results at lines 170-171.

Discussion:

- Can you include your opinion on measures of colon ptosis in relation to your total colon or transverse colon estimation? A very recent study has shown the correlation between slow transit constipation and transverse colon ptosis as measured on abdominal x-rays: Sun et al; Degree of transverse colon ptosis: an alternative surrogate for evaluation of slow transit constipation. Quantitative Imaging in Medicine and Surgery; Vol 13, No 9 (September 01, 2023)

Answer: thank you for this comment, we have added this interesting reference and a comment to this effect in Discussion at Lines 287-290. In these participants we did not encounter substantial colon ptosis and further, larger studies will need to be carried out to investigate this.

- In line 232 you mention that colon length was correlated with age in the FC group only. Could that be due to the fact that the control group was older than the FC group, meaning that many of the controls might have reached the full-grown length of the colon, why a correlation only exists when you are still growing?

Answer: Thank you for the comment, we have added a sentence to this effect in Discussion at lines 237-239. 

Review Comments to the Author: Reviewer #2

1. The authors should provide information concerning the reliability of MRI images

using 3D image analysis skeletonization technique in measurement of colon length.

Answer: Although the paper mentioned this at the beginning of the Results, we entirely agree that this point was not made clearly in Discussion. As such, and also in answer to a similar point raised by Reviewer #1, we have added in Discussion a section to this effect at Lines 259-270. 

2. It is well accepted that MRI study has a limitation in young children who are not cooperative with this time consuming technique. It is noted that the children enrolled in this study aged over 10 years, therefore, the data of this study would be applicable to only older children , not children in general.

Answer: agreed. We have inserted a comment detailing this point and the reasons for the 7 years old cut-off in the inclusion criteria at the end of Discussion Lines 294-297. 

3. Data in this study shows that the colon length of constipated children is longer than healthy children. However, the results do not clearly indicate that colonic length contributes to the pathogenesis of functional constipation. To investigate whether the longer colon is a cause of functional constipation or secondary to prolonged fecal retention , a study should be designed to compare colonic length measured by MRI studies in the same patient at pre treatment and at recovery post treatment of functional constipation.

Answer: thank you for this, it is an important point and we have inserted the comment towards the end of Dicsussion Lines 278-281.

---

## [Editor Report · Decision Letter 1]

1 Dec 2023

PONE-D-23-27259R1Colon length in pediatric health and constipation measured using magnetic resonance imaging and three dimensional skeletonizationPLOS ONE

Dear Dr. Marciani,

Thank you for submitting your manuscript to PLOS ONE. After careful consideration, we feel that it has merit but does not fully meet PLOS ONE’s publication criteria as it currently stands. Therefore, we invite you to submit a revised version of the manuscript that addresses the points raised during the review process.

We look forward to receiving your revised manuscript.

Kind regards,

Tsutomu Kumamoto

Academic Editor

PLOS ONE

Journal Requirements:

Additional Editor Comments:

I believe that the authors have addressed the reviewers’ comments. However, as can be seen from this revision, this study has several limitations. Considering these, it may be necessary to rephrase lines 302 to 305 in the conclusion. Also, please consider whether it would be better to modify the content of the limitations and change where they are mentioned.

---

## [Author Response · Author response to Decision Letter 1]

8 Dec 2023

PONE-D-23-27259 Revision R2

Colon length in pediatric health and constipation measured using magnetic resonance imaging and three dimensional skeletonization

Response to the Editor

Thank you for your comments. We have addressed all the points raised as detailed below in this point-by-point response. All changes made to the manuscript are tracked in the tracked changes Word version enclosed. 

We hope that the manuscript would now meet the journal’s criteria.

Journal Requirements Comments:

Answer: We checked each reference individually as recommended and spotted an incorrect title and some typos in References 4, 10, 21 and 22 which are now corrected. Apologies for not seeing these before.

Additional Editor Comments:

I believe that the authors have addressed the reviewers’ comments. 

Answer: Thank you.

However, as can be seen from this revision, this study has several limitations. Considering these, it may be necessary to rephrase lines 302 to 305 in the conclusion. 

Answer: Agreed. We have rewritten that conclusion to stress that whilst providing new data that will help improve our understanding of colon pathophysiology and drug release modelling in children, the study also highlighted some fundamental limitations of the skeletonization technique that will need addressing to make the method more applicable in the future.

Also, please consider whether it would be better to modify the content of the limitations and change where they are mentioned.

Answer: We reconsidered the location of the limitations paragraph in the Discussion which appeared half-way through the more physio-pathological discussion. As such, we have now moved that section towards the end of the Discussion where such consideration are more traditionally placed. The content of the limitations was already quite comprehensive especially following the comments of the previous technical Reviewer. Accordingly we left the content unchanged.

Figure Upload Comments:

Answer: We run again the three figure files through PACE, which found no image problems.

---

## [Editor Report · Decision Letter 2]

10 Dec 2023

Colon length in pediatric health and constipation measured using magnetic resonance imaging and three dimensional skeletonization

PONE-D-23-27259R2

Dear Dr. Marciani,

We’re pleased to inform you that your manuscript has been judged scientifically suitable for publication and will be formally accepted for publication once it meets all outstanding technical requirements.

Kind regards,

Tsutomu Kumamoto

Academic Editor

PLOS ONE

Additional Editor Comments (optional):

I believe that this study should be accepted by PLOS ONE as the authors have sufficiently addressed the reviewers’ queries.

I hope that this research will contribute to further advancements in this field.
---

## [Editor Report · Acceptance letter]

19 Dec 2023

PONE-D-23-27259R2 

PLOS ONE

Dear Dr. Marciani, 

I'm pleased to inform you that your manuscript has been deemed suitable for publication in PLOS ONE. Congratulations! Your manuscript is now being handed over to our production team.

Kind regards, 

on behalf of

M.D., Ph.D. Tsutomu Kumamoto 

Academic Editor

PLOS ONE